# DDTNet: Degradation Disentanglement and Transfer Network for Domain-Adaptive All-in-One Image De-weathering

## Abstract

All-in-one adverse weather image restoration aims to remove multiple degradations, such as rain, haze, and snow, using a single model. Despite their broad applicability, existing methods typically compromise performance, delivering balanced rather than optimal results for individual degradations due to their multi-task nature. Moreover, they often suffer from a significant performance drop when a domain gap exists between training and testing data. To address these challenges, we propose the Degradation Disentanglement and Transfer Network (DDTNet), which carries out domain adaptation for all-in-one models. Since paired degraded-clean images are unavailable at inference, DDTNet disentangles and transfers degradation patterns from target-domain degraded images to source-domain clean images, generating domain-adaptive pairs for fine-tuning and improving target-specific restoration. The core of DDTNet is the Degradation Disentanglement Module (DDM), which consists of Degradation Coupled Attention (DCA) to capture both general and weather-specific features, enabling effective disentanglement and transfer of degradation patterns. Experimental results demonstrate that DDTNet significantly improves existing all-in-one models across real-world deraining, desnowing, and dehazing datasets.

## 1 Introduction

Image de-weathering seeks to recover clean images from weather-degraded inputs. In the past, much research focuses on addressing a single weather condition, such as deraining (Jiang et al., 2022; Hu et al., 2019; Wang et al., 2020; 2019; Fu et al., 2017; Jiang et al., 2020; Li et al., 2019b; Chen et al., 2024; Gao et al., 2024), dehazing (Wu et al., 2021; Guo et al., 2022; Song et al., 2023; Liu et al., 2019; Deng et al., 2020; Yu et al., 2022; Qin et al., 2020; Zhang et al., 2024; Fang et al., 2025), and desnowing (Chen et al., 2020; 2021; Liu et al., 2018; Zhang et al., 2021; Wang et al., 2017). However, as weather conditions are inherently unpredictable and vary over time, methods designed for a single type of weather-induced degradation have limited practical applicability.

To overcome these limitations, all-in-one image de-weathering (Li et al., 2022; Valanarasu et al., 2022; Chen et al., 2022; Park et al., 2023; Potlapalli et al., 2023; Cui et al., 2025; Sun et al., 2024; Zhang et al., 2023b) has recently gained popularity. Unlike task-specific approaches, a unified model aims to handle multiple weather scenarios within a single framework, making it more adaptable to real-world applications. To achieve effective all-in-one restoration, several studies have incorporated weather-specific prompts, such as weather type queries (Valanarasu et al., 2022) and degradation-specific prompts (Potlapalli et al., 2023; Tian et al., 2025). However, despite enhancing the model's adaptability, prompt-based methods still face two major issues. First, the joint multi-task training strategy often results in suboptimal performance on individual tasks, with a noticeable performance drop compared to single-task models. Second, most all-in-one methods suffer from a domain gap between training and testing data, which limits generalization ability and causes ineffectiveness in real-world deployment.

To address the challenges of cross-task performance degradation and domain sensitivity, we introduce a Degradation Disentanglement and Transfer Network (DDTNet) for domain adaptation in all-in-one image de-weathering.

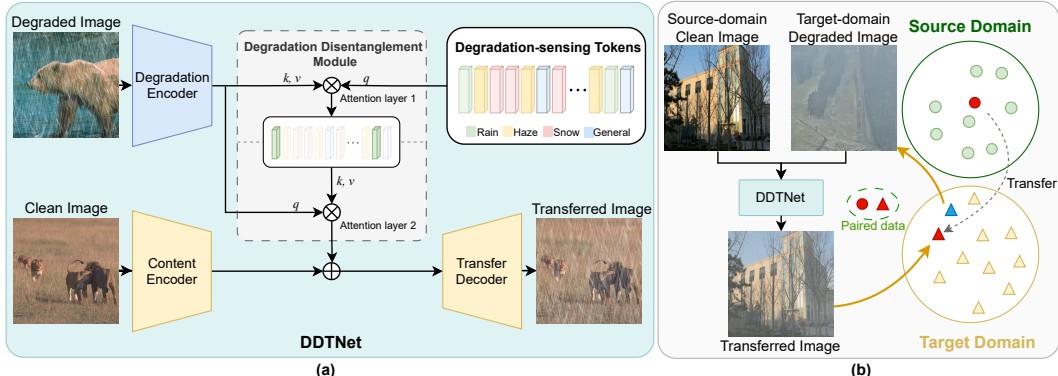

Figure 1: (a) For test-time adaptation, DDTNet disentangles and transfers degradation patterns from a target-domain degraded image to a source-domain clean image, generating paired images for fine-tuning an arbitrary all-in-one restoration model. (b) DDTNet comprises the Degradation Disentanglement Module (DDM), where a two-stage Degradation Coupled Attention (DCA) is developed to separate degradation patterns from a degraded image. The separated degradation patterns are then transferred via an encoder-decoder architecture.

As shown in Figure 1(a), the key component of DDTNet is the Degradation Disentanglement Module (DDM), which uses a Degradation Coupled Attention (DCA) mechanism to effectively disentangle diverse degradation patterns. The DCA is a two-stage attention mechanism applied to learnable *degradation-sensing tokens*. In the first stage, these tokens serve as queries to retrieve degraded features from the input image, yielding *degradation tokens* that encode the underlying degradation patterns. In the second stage, the roles are reversed: the image tokens act as queries to aggregate the information stored in these new degradation tokens for distilling degradation features. Repeating this two-stage interaction can progressively disentangle degradation patterns from the image. This disentanglement strategy is the core of our work, which allows DDTNet to transfer isolated degradation patterns from a target-domain degraded image to a source-domain clean image. As shown in Figure 1(b), this transfer process generates domain-adaptive paired data, consisting of clean source images with target-domain degradation patterns. These synthetic yet aligned pairs not only embed target-domain characteristics but also adapt to weather-specific conditions such as rain, haze, snow, or even mixed weather, which are then used to fine-tune de-weathering models. Through explicitly aligning the restoration with target-domain degradations, DDTNet improves model performance and restoration quality in challenging real-world scenarios.

This work makes three primary contributions. First, we present DDTNet to tackle two critical challenges in all-in-one image restoration: suboptimal cross-task performance and sensitivity to domain gaps. DDTNet disentangles and transfers degradation patterns from target-domain degraded images to source-domain clean images, hence generating paired, domain-adaptive images for fine-tuning all-in-one restoration models. Second, we propose DCA, a two-stage attention mechanism that effectively identifies and separates degradation patterns from degraded images by alternating the roles of queries and key-value pairs between degradation sensing tokens and image tokens. Finally, experimental results show that DDTNet significantly improves existing all-in-one restoration models on benchmark real-world deraining, desnowing, and dehazing datasets.

## 2 RELATED WORK

**All-in-One Image Restoration.** All-in-one image restoration aims to address multiple degradations using a unified model. To tackle this task, several studies (Chen et al., 2022; Zhu et al., 2023; Potlapalli et al., 2023; Cui et al., 2025; Tian et al., 2025) learn both degradation-specific and degradation-agnostic features within a unified framework. Potlapalli et al. (2023) integrate degradation-specific cues into a unified model via learnable prompts to handle diverse degradations. Cui et al. (2025) extract degradation-specific frequency subbands to adaptively address different degradations through frequency mining and modulation. Tian et al. (2025) introduce degradation-aware feature perturbations to align degradation-specific prompts with the unified model. Although these methods demonstrate the potential of unified networks to handle diverse degradations, they often yield suboptimal performance on individual tasks due to the joint multi-task training strategy. Additionally, the do-

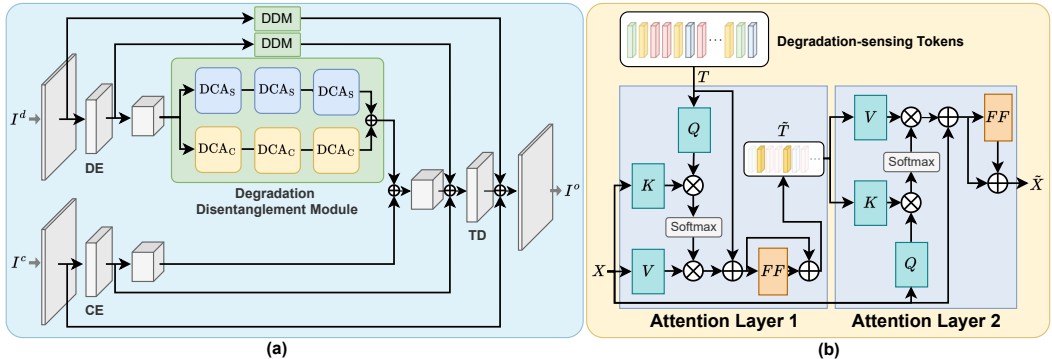

Figure 2: (a) DDTNet is a dual-branch network whose top branch extracts degradation features from a degraded image $I^d$ via a Degradation Encoder (DE) and a Degradation Disentanglement Module (DDM). The DDM leverages Spatial-wise and Channel-wise Degradation Coupled Attention (S-DCA and C-DCA) to effectively separate degradation features. In contrast, the bottom branch extracts content features from a clean image $I^c$ using a Content Encoder (CE). Finally, it fuses degradation and content features to generate the degradation-transferred image $I^o$. (b) DCA is a two-stage attention mechanism. In the first stage, degradation-sensing tokens ($T$) retrieve degraded features from the input image to produce degradation tokens ($\tilde{T}$) that encode the underlying degradation patterns. In the second stage, the image tokens act as queries to aggregate the information stored in $\tilde{T}$, thereby distilling degradation features.

main gap between training and testing data frequently causes significant performance drops, severely degrading restoration results in real-world scenarios.

**Domain Adaptation for Image Restoration.** Domain adaptation in the field of image restoration aims to narrow the domain gap between the source domain and the target domains, ensuring that models trained on synthetic data generalize well to real-world scenarios. There were several early studies (Wei et al., 2021; Shao et al., 2020) focusing on single-degradation scenarios. For instance, Wei et al. (2021) and Shao et al. (2020) utilize CycleGAN Zhu et al. (2017) to generate pseudo-training data for deraining and dehazing tasks. Chi et al. (2021) introduce a meta-auxiliary learning strategy to enable fast test-time adaptation for deblurring. Although these methods alleviate the domain gap issue in single-degradation scenarios, they lack the flexibility to generalize across multiple types of degradations. Therefore, Liao et al. (2025) propose a general domain adaptation framework built upon a pre-trained diffusion model (Ho et al., 2020), which computes a diffusion loss to align restored outputs of synthetic and real-world data. However, since this method is not tailored for all-in-one image restoration, which requires both degradation-specific and degradation-general features, it cannot effectively handle multiple degradations within a unified framework. In contrast, our proposed DDTNet integrates both degradation-specific and degradation-general features to achieve degradation-transfer–driven adaptation, which transfers degradation patterns from target domains to clean images in the source domain and, in turn, enables robust domain adaptation across diverse restoration tasks.

## 3 PROPOSED METHOD

### 3.1 OVERVIEW

This section introduces the Degradation Disentanglement and Transfer Network (DDTNet), a framework designed to transfer degradation patterns from degraded images in an unseen target domain to clean images in a source domain. Synthesizing domain-adaptive image pairs, DDTNet facilitates test-time adaptation via this transfer process. It enables fine-tuning and achieves performance improvement of restoration models under unseen degradation conditions.

As illustrated in Figure 2(a), DDTNet consists of two parallel branches: the top branch for extracting degradation features and the bottom branch for extracting content features. Given a degraded image $I^d \in \mathbb{R}^{H \times W \times 3}$ and a clean image $I^c \in \mathbb{R}^{H \times W \times 3}$, DDTNet employs a Degradation Encoder (DE) and a Content Encoder (CE) to obtain multi-scale degradation features

$\{F_i{}^d \in \mathbb{R}^{\frac{H}{2^i} \times \frac{W}{2^i} \times C_i}\} = \mathrm{DE}(I^d)$ and content features $\{F_i{}^c \in \mathbb{R}^{\frac{H}{2^i} \times \frac{W}{2^i} \times C_i}\} = \mathrm{CE}(I^c)$, where the channel dimension $C_i = 32 \cdot 2^i$, and $i \in \{0, 1, 2\}$ indexes the feature scale. At each scale, the degradation features $F_i{}^d$ are processed by the proposed Degradation Disentanglement Module (DDM), which integrates three parallel Spatial-wise Degradation-Coupled Attention ($\mathrm{DCA_S}$) and Channel-wise Degradation-Coupled Attention ($\mathrm{DCA_C}$) layers. The DDM disentangles degradation patterns to produce distilled degradation features $\tilde{F}_i^d = \mathrm{DDM}(F_i^d) \in \mathbb{R}^{\frac{H}{2^i} \times \frac{W}{2^i} \times C_i}$, effectively suppressing scene content while preserving degradation cues as

$$\mathrm{DDM}(F_i^d) = \mathrm{Conv}\left(\mathrm{Concat}\left((\mathrm{DCA_S})^3(F_i^d), (\mathrm{DCA_C})^3(F_i^d)\right)\right), \tag{1}$$

where $(\mathrm{DCA_S})^3$ and $(\mathrm{DCA_C})^3$ denote the sequential application of three spatial and three channel attention layers, respectively.

We apply the Transfer Decoder (TD) to fuse multi-scale distilled degraded features $\tilde{F}_i^d$ and content features $F_i^c$, generating the degradation-transferred image $I^o \in \mathbb{R}^{H \times W \times 3}$ as

$$F^i = \mathrm{TD}_i\left(\mathrm{Concat}\left(\tilde{F}_i^d, F_i^c, F^{(i+1)\uparrow}\right)\right), \quad i \in \{0, 1, 2\}, \tag{2}$$

where $F^3 := \varnothing$, and $F^0 := I^0$. Here, $(\cdot)^\uparrow$ denotes upsampling by a factor of two, and $\{\mathrm{TD}_i\}_{i=0}^2$ are the Transfer Decoders for different scales. In the following, we detail the DCA module and its spatial and channel variants $\mathrm{DCA_S}$ and $\mathrm{DCA_C}$.

## 3.2 Degradation Coupled Attention (DCA)

The goal of our proposed DCA is to disentangle degradation patterns not only in single-degradation scenarios but also under complex mixed degradations, such as rain combined with haze or snow combined with haze. As illustrated in Figure 2(b), DCA adopts a two-stage attention mechanism guided by degradation-sensing tokens, enabling it to capture both degradation-general and degradation-specific features.

Let the input features of DCA be $X \in \mathbb{R}^{N \times D}$, which contains $N$ tokens of dimension $D$. In the first stage, $X$ is processed by linear projection layers $\mathbf{L}$ to generate the key and value features, i.e., $X^k \in \mathbb{R}^{N \times D}$ and $X^v \in \mathbb{R}^{N \times D}$, which is formulated as

$$(X^k, X^v) = \mathbf{L}(X). \tag{3}$$

We then introduce a set of learnable degradation-sensing tokens $T \in \mathbb{R}^{M \times D}$, consisting of $M$ tokens of dimension $D$. These tokens are linearly projected as queries, $T^q = \mathbf{L}(T)$, to retrieve degradation-related information from $X$, which produces the degradation tokens $\tilde{T} \in \mathbb{R}^{M \times D}$. The resulting $\tilde{T}$ can be regarded as a set of degradation kernels that encode the underlying degradation patterns. This process is carried out via cross attention, namely,

$$\tilde{T} = \mathrm{softmax}(\frac{T^q \cdot (X^k)^\top}{\sqrt{D}}) \cdot X^v + T, \tag{4}$$

where $T^q = \mathbf{L}(T)$, and $\top$ denotes the transpose operation. In practice, we set $M = 256$.

In the second stage, the roles are reversed: the input features $X$ act as queries to aggregate the information stored in the degradation tokens $\tilde{T}$, thereby distilling degradation features. Specifically, we process $X$ and $\tilde{T}$ through linear projection layers $\mathbf{L}$ to generate the query, key, and value features, denoted as $X^q \in \mathbb{R}^{N \times D}$, $\tilde{T}^k \in \mathbb{R}^{M \times D}$, and $\tilde{T}^v \in \mathbb{R}^{M \times D}$, respectively, as

$$X^q = \mathbf{L}(X) \quad \text{and} \quad (\tilde{T}^k, \tilde{T}^v) = \mathbf{L}(\tilde{T}). \tag{5}$$

Next, the distilled degradation features $\tilde{X} \in \mathbb{R}^{N \times D}$, which retain only degradation cues while suppressing scene content, are obtained via

$$\tilde{X} = \mathrm{softmax}(\frac{X^q \cdot (\tilde{T}^k)^\top}{\sqrt{D}}) \cdot \tilde{T}^v + X. \tag{6}$$

To effectively disentangle degradation patterns in the proposed DDM, we apply DCA along both the spatial and channel dimensions of the features. The Spatial-wise DCA ($\mathrm{DCA_S}$) emphasizes the geometric structure and spatial distribution of degradations, while the Channel-wise DCA ($\mathrm{DCA_C}$) concentrates on the contrast and intensity attenuation of degradation, as detailed below.

**Spatial-wise Degradation Coupled Attention** ($\text{DCA}_\text{S}$). Let the input to $\text{DCA}_\text{S}$ be $X \in \mathbb{R}^{H \times W \times C}$. We first reshape $X$ into a 2D tensor $X \in \mathbb{R}^{HW \times C}$, corresponding to $HW$ tokens of $C$ dimensions. The reshaped $X$ is then processed by the DCA operations defined in Equations (3)–(6). Lastly, the output is reshaped back to $H \times W \times C$ to produce the result of $\text{DCA}_\text{S}$.

**Channel-wise Degradation Coupled Attention** ($\text{DCA}_\text{C}$). Let the input to $\text{DCA}_\text{C}$ be $X \in \mathbb{R}^{H \times W \times C}$. We first apply spatial pooling to $X$ to obtain $X_\text{pool} \in \mathbb{R}^{24 \times 24 \times C}$, and then reshape it into a 2D tensor $X_\text{pool} \in \mathbb{R}^{C \times 576}$, corresponding to $C$ tokens of dimension 576. The pooled features are then processed by the DCA operations defined in Equations (3)–(6), and the output is reshaped back to $24 \times 24 \times C$. Finally, bilinear upsampling is utilized to restore the resolution to $H \times W \times C$, yielding the output of $\text{DCA}_\text{C}$.

### 3.3 LOSS FUNCTION

DDTNet is trained on degraded–clean-mixed triplets $\{(I_i^d,\, I_i^c,\, I_i^m)\}_{i=1}^N$, where the ground-truth mixed image $I_i^m$ preserves the scene content of $I_i^c$ while exhibiting the same degradation patterns as $I_i^d$, as described in Section 4.1. To supervise the generation of each degradation-transferred image $I_i^o = \text{DDTNet}(I_i^d, I_i^c)$, we adopt the $\ell_1$ reconstruction loss, defined as

$$\mathcal{L} = \frac{1}{N} \sum_{i=1}^N \|I_i^o - I_i^m\|_1. \tag{7}$$

### 3.4 DOMAIN-ADAPTIVE FINE-TUNING PROCESS

After training, DDTNet is employed to transfer degradation patterns from target-domain degraded images $\{I_i^d\}_{i=1}^N$ to source-domain clean images $\{I_i^c\}_{i=1}^N$, randomly sampled from a source-domain dataset. As illustrated in Figure 1(b), for each pair $(I_i^d, I_i^c)$, DDTNet generates a degradation-transferred image $I_i^o = \text{DDTNet}(I_i^d, I_i^c)$, which preserves the scene content of $I_i^c$ while embedding the degradation patterns from $I_i^d$. This process allows us to construct domain-adaptive training pairs $\mathcal{D}_\text{adapt} = \{(I_i^o, I_i^c)\}_{i=1}^N$, which are then used to update restoration models during testing, thereby enhancing their generalization and performance in the target domain.

## 4 EXPERIMENTS

### 4.1 IMPLEMENTATION DETAILS.

**Datasets.** We construct a set of degraded–clean–mixed triplets $\{(I_i^d,\, I_i^c,\, I_i^m)\}_{i=1}^N$, where $I_i^d$ denotes a degraded image, $I_i^c$ a clean image, and $I_i^m$ a synthesized mixed image that preserves the scene content of $I_i^c$ while incorporating the degradation patterns of $I_i^d$. To generate $I_i^m$ with consistent degradation patterns but diverse scene contents, we employ rain masks from Rain100H and Rain100L (Yang et al., 2017), as well as snow masks from Snow100K (Liu et al., 2018), to synthesize rainy and snowy images. For hazy images, we directly adopt RESIDE (Li et al., 2019a), which provides hazy images with controlled haze density and atmospheric light across different scenes.

During training, we sample $5,000$ triplets for each of the three tasks, yielding a total of $15,000$ pairs for jointly optimizing DDTNet and the restoration models. To evaluate the effectiveness of DDTNet, we use the real-world WeatherStream dataset (Zhang et al., 2023a), which contains $4,500$ hazy, $3,000$ rainy, and $3,960$ snowy images with corresponding clean counterparts. This dataset poses a particularly challenging benchmark as it includes not only single-degraded cases but also mixed degradations, such as rain with haze and snow with haze.

**DDTNet Configuration.** We optimize DDTNet using the Adam optimizer with a learning rate of $1 \times 10^{-4}$, a batch size of 4, and 150 training epochs. DDTNet comprises 31 million parameters and achieves an inference time of 33 ms on an NVIDIA RTX A5000 GPU, with inputs resized to $256 \times 256$.

**Restoration Models.** To evaluate the effectiveness of DDTNet for domain adaptation, we adopt three state-of-the-art (SOTA) restoration models: PromptIR (Potlapalli et al., 2023), AdaIR (Cui et al., 2025), and DFPIR (Tian et al., 2025). All models are initially trained in an all-in-one manner

Table 1: Performance gains with DDTNet on three restoration models, PromptIR, AdaIR, and DFPIR on WeatherStream across three real-world weather types: rain, snow, and haze.

| | Method | Rain PSNR ↑ | Rain SSIM ↑ | Snow PSNR ↑ | Snow SSIM ↑ | Haze PSNR ↑ | Haze SSIM ↑ | Average PSNR ↑ | Average SSIM ↑ |
|---|---|---|---|---|---|---|---|---|---|
| **PromptIR** | Baseline | 22.96 | 0.762 | 21.36 | 0.738 | 19.08 | 0.689 | 21.13 | 0.730 |
| | +DDTNet | **23.82** (+0.86) | **0.770** (+0.008) | **22.12** (+0.76) | **0.744** (+0.006) | **20.72** (+1.64) | **0.695** (+0.006) | **22.22** (+1.09) | **0.736** (+0.006) |
| **AdaIR** | Baseline | 22.92 | 0.761 | 21.26 | 0.738 | 18.58 | 0.687 | 20.92 | 0.729 |
| | +DDTNet | **24.17** (+1.25) | **0.775** (+0.014) | **22.49** (+1.23) | **0.749** (+0.011) | **20.48** (+1.90) | **0.701** (+0.014) | **22.36** (+1.44) | **0.742** (+0.013) |
| **DFPIR** | Baseline | 22.95 | 0.753 | 21.12 | 0.717 | 20.09 | 0.675 | 21.19 | 0.710 |
| | +DDTNet | **24.04** (+1.09) | **0.765** (+0.012) | **21.80** (+0.68) | **0.719** (+0.002) | **21.16** (+1.07) | **0.680** (+0.005) | **22.13** (+0.94) | **0.716** (+0.006) |
| **Average Gain** | | **+1.07** | **+0.011** | **+0.89** | **+0.006** | **+1.54** | **+0.008** | **+1.16** | **+0.008** |

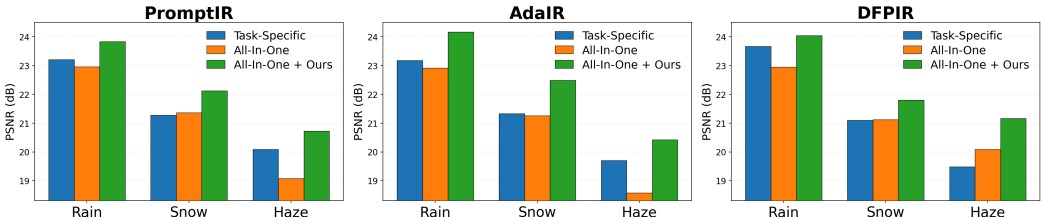

Figure 3: Quantitative comparison of training schemes. All-in-one methods (PromptIR, AdaIR, and DFPIR) often underperform compared to their task-specific versions. In contrast, DDTNet enhances the performance of all-in-one models, allowing them to surpass task-specific performance.

on the 15,000 synthesized training pairs, following their respective default training configurations. During testing, each model is further fine-tuned for a single epoch using the domain-adaptive pairs generated by DDTNet, enabling efficient adaptation to the target domain.

## 4.2 Performance Evaluations

**Quantitative Comparison.** Table 1 reports a quantitative comparison of three SOTA all-in-one image restoration models: PromptIR (Potlapalli et al., 2023), AdaIR (Cui et al., 2025), and DFPIR (Tian et al., 2025). Here, "Baseline" denotes models trained without DDTNet, while "+DDTNet" refers to their DDTNet-enhanced counterparts. The results clearly show that DDTNet consistently and significantly boosts performance across all models on WeatherStream. Specifically, our DDTNet yields an impressive average PSNR gain of 1.09 dB for PromptIR, 1.44 dB for AdaIR, and 0.94 dB for DFPIR. Furthermore, when broken down by task, DDTNet achieves average PSNR improvements of 1.07 dB for deraining, 0.89 dB for desnowing, and 1.54 dB for dehazing.

Figure 3 further compares the three restoration models under three training schemes: (1) task-specific training, (2) all-in-one training, and (3) all-in-one training enhanced with DDTNet. The results confirm that all-in-one training typically suffers from inter-task interference, yielding suboptimal results compared to task-specific training. In contrast, integrating DDTNet not only alleviates this interference but also enables all-in-one models to outperform their task-specific counterparts. Overall, these findings show that DDTNet not only improves restoration performance on unseen target domains but also effectively mitigates the limitations of conventional all-in-one training schemes.

**Qualitative Comparison.** Figure 4 illustrates several examples of degradation-transferred images, where the degraded inputs are selected from WeatherStream and the clean images are sampled from RESIDE. In each case, DDTNet faithfully reproduces the degradation patterns of the degraded image while preserving the scene content of the clean image. This shows that DDTNet can successfully generate domain-adaptive pairs that align with the target-domain degradation distribution, thereby enabling effective fine-tuning of restoration models and improving their performance during testing.

We further present qualitative comparisons of de-weathering results using three restoration models: PromptIR (Figure 6), AdaIR (Figure 7), and DFPIR (Figure 8). Here, "Baseline" denotes the models trained without DDTNet, whereas "+DDTNet" indicates their DDTNet-enhanced counterparts. The baseline models often struggle to cope with diverse degradation patterns in the target domain,

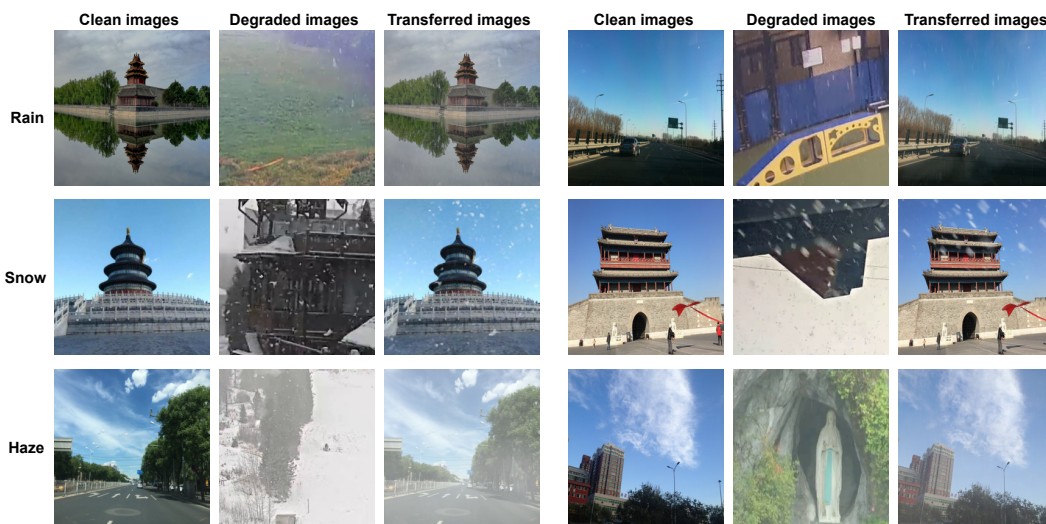

Figure 4: Qualitative results of degradation-transferred images. Degradation patterns from WeatherStream target-domain images are transferred to clean source-domain images from RESIDE.

while the DDTNet-enhanced models effectively remove complex degradations, producing visually superior and perceptually more pleasing results. These visualizations highlight DDTNet's strong cross-domain generalization capability in handling diverse real-world weather degradations.

**Analysis of Degradation Tokens.** Figure 5 presents a t-SNE visualization of the learned degradation tokens $\tilde{T}$, obtained by randomly sampling 50 images per task (deraining, desnowing, and dehazing). It reveals that the degradation tokens are well-clustered according to weather type: rain samples (blue) are grouped in the upper-right, haze samples (orange) in the lower-left, and snow samples (green) near the center. This clear separation across weather types, combined with strong intra-class consistency, indicates that the degradation tokens $\tilde{T}$ effectively capture and disentangle degradation-specific features. Such discriminative representations are crucial for enabling reliable degradation transfer within an all-in-one framework.

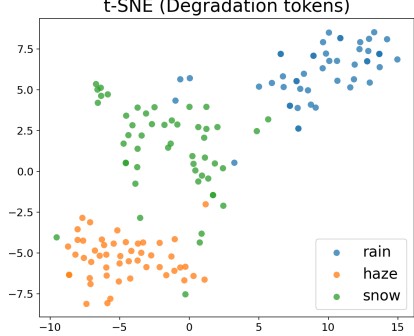

Figure 5: t-SNE visualization of the learned degradation tokens.

**Comparison with Existing Domain Adaptation Methods.** Since most existing domain adaptation methods for image restoration are tailored to single degradations, we compare DDTNet with the only general-purpose domain adaptation approach, Noise-DA (Liao et al., 2025). To ensure a fair comparison, we re-implement Noise-DA with its official restoration backbones and train it on our dataset under the all-in-one training scheme. As reported in Table 2, Noise-DA exhibits limited generalization capabilities: while it yields gains on the deraining and desnowing tasks, it causes a performance drop on dehazing, reflecting its lack of adaptability to diverse degradations in an all-in-one setting. In contrast, DDTNet disentangles heterogeneous degradation patterns to generate domain-adaptive pairs, consistently boosting the restoration performance across all tasks and achieving a substantial average PSNR gain of 5.67 dB over the baseline.

## 4.3 ABLATION STUDIES

To analyze the quality of the generated degradation-transferred images, we construct a test set of 900 degraded–clean-mixed triplets $\{(I_i^d, I_i^c, I_i^m)\}_{i=1}^{900}$ on Rain100H/Rain100L (Yang et al., 2017), Snow100K (Liu et al., 2018), and RESIDE (Li et al., 2019a), with 300 triplets per task, to synthesize rainy and snowy images. We then provide a component analysis of DDTNet and DCA.

**Ablation Study of DDTNet.** Table 3 presents an ablation study of DDTNet with its two DCA components: $DCA_S$ and $DCA_C$. We implement four variants of the network, including 1) Net1: a pure encoder–decoder architecture without DDM, serving as the baseline; 2) Net2: Net1 augmented

Table 2: Quantitative comparison between DDTNet and Noise-DA on WeatherStream, with the restoration backbone employed by Noise-DA as the baseline.

| Method | Rain PSNR ↑ | Rain SSIM ↑ | Snow PSNR ↑ | Snow SSIM ↑ | Haze PSNR ↑ | Haze SSIM ↑ | Average PSNR ↑ | Average SSIM ↑ |
|---|---|---|---|---|---|---|---|---|
| Baseline | 16.23 | 0.496 | 15.32 | 0.484 | 18.06 | 0.561 | 16.74 | 0.514 |
| +Noise-DA | 19.11 (+2.88) | 0.596 (+0.100) | 17.24 (+1.92) | 0.577 (+0.093) | 16.93 (-1.13) | 0.583 (+0.022) | 17.76 (+1.02) | 0.586 (+0.072) |
| **+DDTNet** | **24.56** (+8.33) | **0.779** (+0.283) | **21.55** (+6.23) | **0.735** (+0.251) | **21.12** (+3.06) | **0.699** (+0.138) | **22.41** (+5.67) | **0.737** (+0.223) |

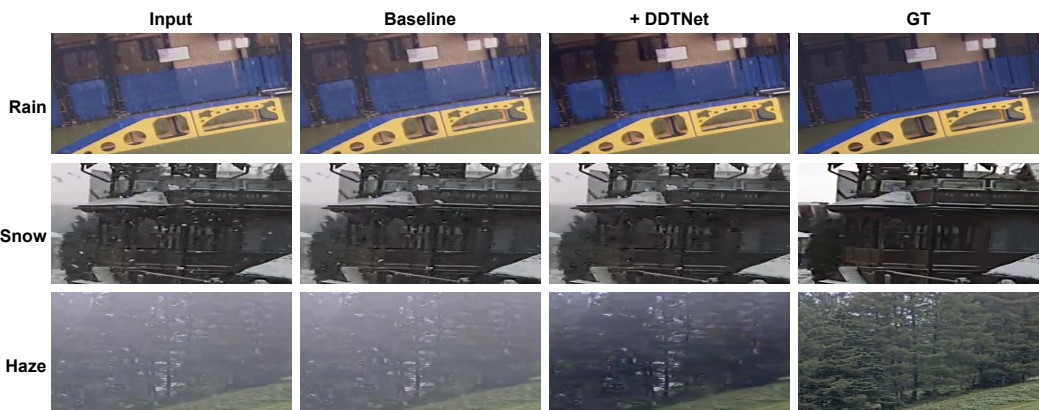

Figure 6: Qualitative comparison of PromptIR Potlapalli et al. (2023) on WeatherStream between its baseline and DDTNet-enhanced versions.

Table 3: Component analysis of DDTNet on the average PSNR of degradation-transferred images across Rain100H/L, Snow100K, and RESIDE.

| | Enc–Dec | DDM DCA$_S$ | DCA$_C$ | PSNR |
|---|---|---|---|---|
| Net1 | ✓ | | | 31.59 |
| Net2 | ✓ | ✓ | | 34.30 |
| Net3 | ✓ | | ✓ | 32.44 |
| Ours | ✓ | ✓ | ✓ | **34.53** |

Table 4: Component analysis of Degradation-Coupled Attention (DCA) on the average PSNR of degradation-transferred images across Rain100H/L, Snow100K, and RESIDE.

| | Enc–Dec | Degr.-Sensing Tokens | Attn–L1 | Attn–L2 | PSNR |
|---|---|---|---|---|---|
| Net1 | ✓ | | | | 31.59 |
| Net4 | ✓ | | ✓ | ✓ | 33.00 |
| Net5 | ✓ | ✓ | | ✓ | 33.35 |
| Ours | ✓ | ✓ | ✓ | ✓ | **34.53** |

with DCA$_S$ alone in the DDM; 3) Net3: Net1 augmented with DCA$_C$ alone in the DDM; and 4) Ours: the complete DDTNet with both DCA$_S$ and DCA$_C$. The results show that Net2 consistently surpasses Net1 across all three tasks, while Net3 enhances Net1 on deraining and dehazing but not on desnowing. Our proposed DDTNet achieves the best overall performance among all variants. Specifically, DDTNet improves the PSNR of the baseline encoder-decoder model by 2.07 dB for deraining, 1.80 dB for desnowing, and 4.93 dB for dehazing.

**Ablation Study of DCA.** Table 4 analyzes the effectiveness and contributions of the proposed Degradation-Coupled Attention (DCA). DCA is composed of three key components: degradation-sensing tokens, attention layer 1, and attention layer 2. We compare four module configurations, including 1) Net1: a pure encoder–decoder architecture without DDM, identical to the baseline in Table 3; 2) Net4: Net1 augmented with attention layers 1 and 2 in DCA, but without degradation-sensing tokens. In this case, the attention layers directly perform self-attention on the input features, functioning as a purely data-driven mechanism; 3) Net5: Net1 augmented with degradation-sensing tokens and attention layer 2, where the tokens are directly fed into the second attention layer without applying the coupling mechanism; 4) Ours: the full DDTNet with all DCA components. The results demonstrate that both degradation-sensing tokens and the coupled attention layers individually improve the baseline. Crucially, the complete integration of all three components significantly outperforms the other variants. This verifies the effectiveness of the proposed degradation-coupled attention in facilitating robust degradation transfer.

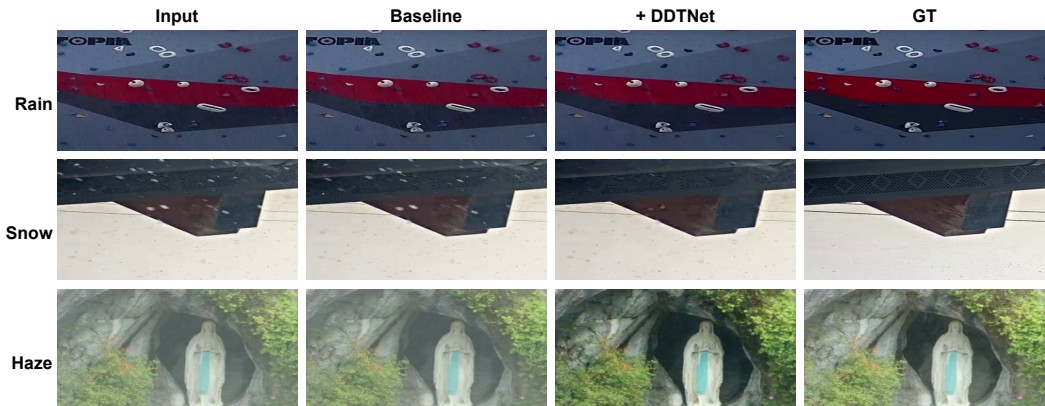

Figure 7: Qualitative comparison of AdaIR on WeatherStream between its baseline and DDTNet-enhanced versions.

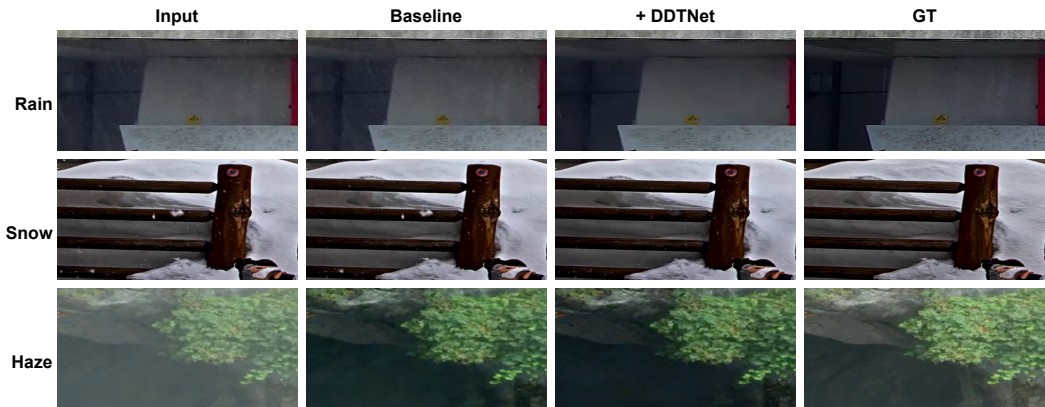

Figure 8: Qualitative comparison of DFPIR on WeatherStream between its baseline and DDTNet-enhanced versions.

**Limitations.** DDTNet requires degraded-clean-mixed triplets for supervision during training, which may limit scalability when such triplets are scarce or costly to obtain.

## 5 CONCLUSION

We propose the Degradation Disentanglement and Transfer Network (DDTNet), a novel approach designed to address domain adaptation in all-in-one image de-weathering. Since paired degraded–clean images are typically unavailable during testing, DDTNet transfers degradation patterns from target-domain degraded images onto source-domain clean images, thereby generating domain-adaptive pairs for test-time fine-tuning of restoration models. This adaptation process can significantly enhance the generalization capability of restoration models to unseen domains. The core of DDTNet lies in the Degradation-Coupled Attention mechanism, which integrates learnable degradation-sensing tokens with a two-stage attention process to disentangle degradation patterns. In the first stage, the degradation-sensing tokens query the input features to encode degradation cues and generate degradation tokens. In the second stage, the input features act as queries to aggregate information from these degradation tokens, thus distilling degradation features. Extensive experiments demonstrate that DDTNet substantially improves the performance of state-of-the-art all-in-one restoration models across real-world deraining, dehazing, and desnowing benchmarks, validating its effectiveness for robust cross-domain adaptation.

## ETHICS STATEMENT

This study develops a model for transferring weather-related degradation patterns. It does not involve human subjects, personal data, or sensitive content, and it adheres to the ICLR Code of Ethics. All experiments use publicly available deraining, desnowing, and dehazing datasets under appropriate licenses. We do not foresee privacy, safety, or fairness concerns; the method is intended solely to improve image quality in adverse-weather scenarios and is not designed for harmful applications.

## REPRODUCIBILITY STATEMENT

All experimental results in this study are reproducible, and the implementation details (Section 3.2) and experimental settings (Section 4.1) are specified in the main text. The full codebase, pretrained weights, and documentation will be released publicly upon acceptance.

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

# A APPENDIX

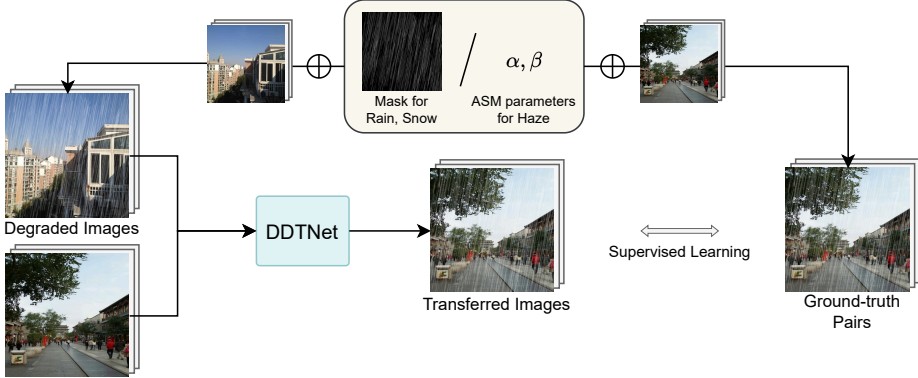

Figure A1: Pipeline of data synthesis and optimization in DDTNet. We synthesize degraded–clean triplets sharing the same degradation but different content, enabling end-to-end optimization of the network.

## A.1 PIPELINE OF DATA SYNTHESIS AND OPTIMIZATION IN DDTNET.

As shown in Figure A1, we collect degraded–clean triplets that share the same degradation patterns but different in content. For rainy and snowy images, we use the degradation masks (rain streaks and snow masks) from Rain100H (Yang et al., 2017), Rain100L (Yang et al., 2017), and Snow100K (Liu et al., 2018) to synthesize the degraded images. For hazy images, we directly adopt the RESIDE dataset (Li et al., 2019a), which provides hazy images sharing the same haze characteristics (haze density and atmospheric light) but differing in content. Next, we describe the data synthesis process for generating rainy, snowy, and hazy images.

**Rain and snow.** Given a clean image $I^c \in \mathbb{R}^{H \times W \times 3}$, we utilize rain or snow masks $M \in \mathbb{R}^{H \times W \times 1} \in [0, 1]$ to generate rainy or snowing image $I^d \in \mathbb{R}^{H \times W \times 3}$ as

$$I^d = (1 - \lambda M) \odot I^c + (\lambda M) \odot \mathbf{c}, \tag{8}$$

where $\odot$ denotes element-wise multiplication, $\lambda \in [0, 1]$ is the mask coefficient, and the $\mathbf{c} \in \mathbb{R}^{1 \times 1 \times 3} \in [0, 1]$ represents the chromatic aberration value.

**Haze.** Given a clean image $I^c \in \mathbb{R}^{H \times W \times 3}$, previous methods (Li et al., 2019a) often rely on the atmospheric scattering model (ASM) to generate hazy image $I^h \in \mathbb{R}^{H \times W \times 3}$ as

$$\begin{aligned} I^h &= I^c \times T + \alpha \times (1 - T), \\ T &= e^{-\beta \times d}, \end{aligned} \tag{9}$$

where $\alpha \in \mathbb{R}^3$ denotes the atmospheric light, $T \in \mathbb{R}^{H \times W \times 1}$ denotes the transmission map, $\beta \in \mathbb{R}^1$ and $d \in \mathbb{R}^{H \times W \times 1}$ denote the haze density and depth map.

## A.2 ADDITIONAL QUALITATIVE RESULTS

Figure A2 presents additional degradation-transferred results along with their corresponding degradation features. These results demonstrate that DDTNet successfully disentangles and transfers degradation patterns, thereby producing realistic degradation-transferred images. Figures A3 to A5 present additional restored images generated by PromptIR (Potlapalli et al., 2023), AdaIR (Cui et al., 2025), and DFPIR (Tian et al., 2025), respectively. These results demonstrate that these methods effectively enhance the baseline model, removing artifacts and producing more realistic images.

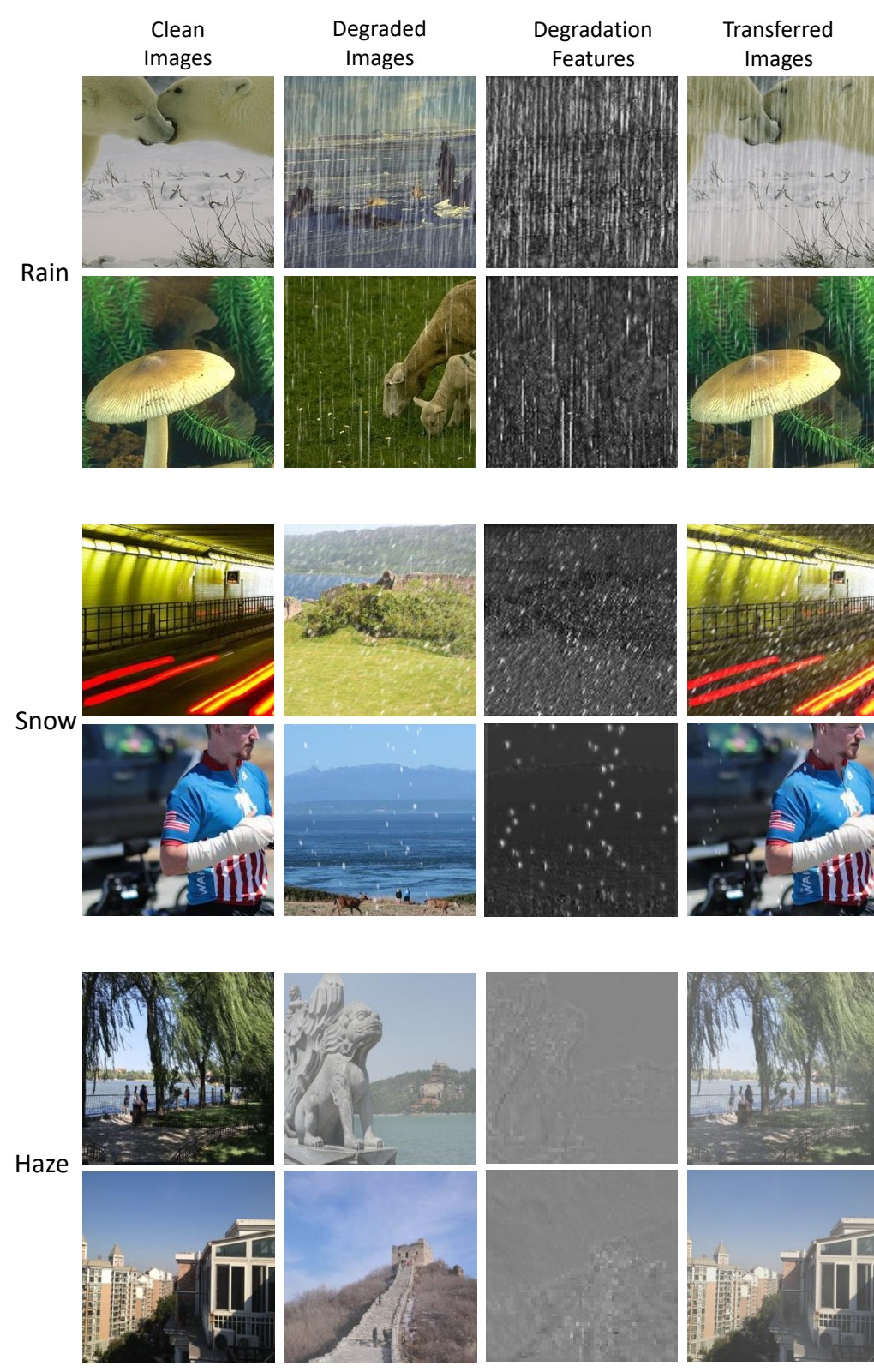

Figure A2: Qualitative results of degradation-transferred images.

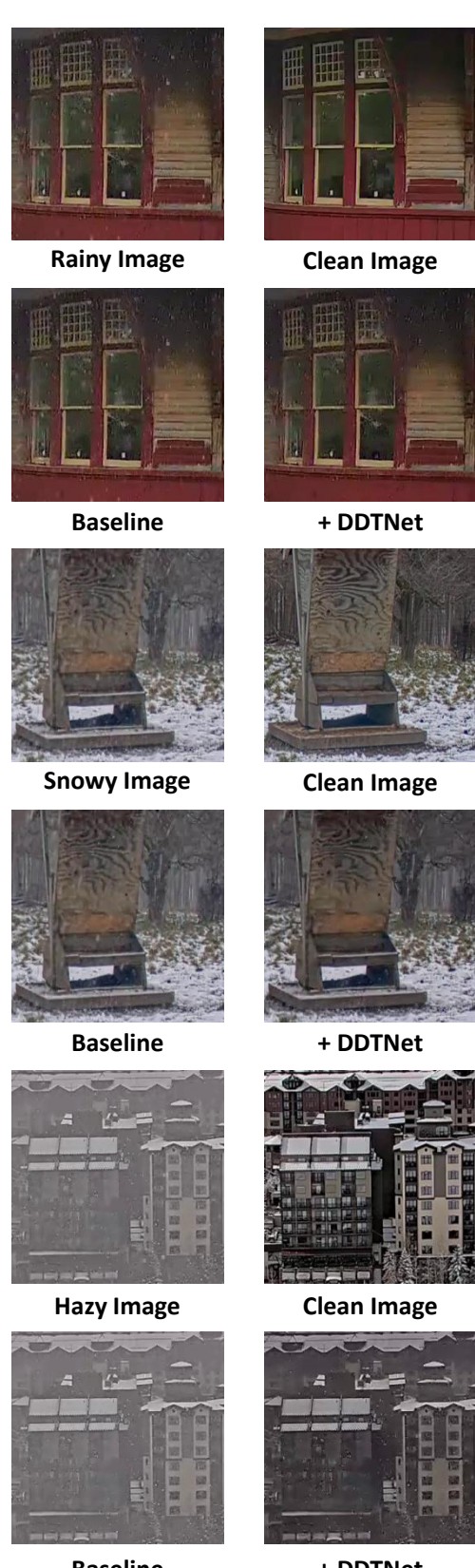

Figure A3: Qualitative comparison of PromptIR Potlapalli et al. (2023) on the Weather-Stream (Zhang et al., 2023a) between its baseline and DDTNet-enhanced versions.

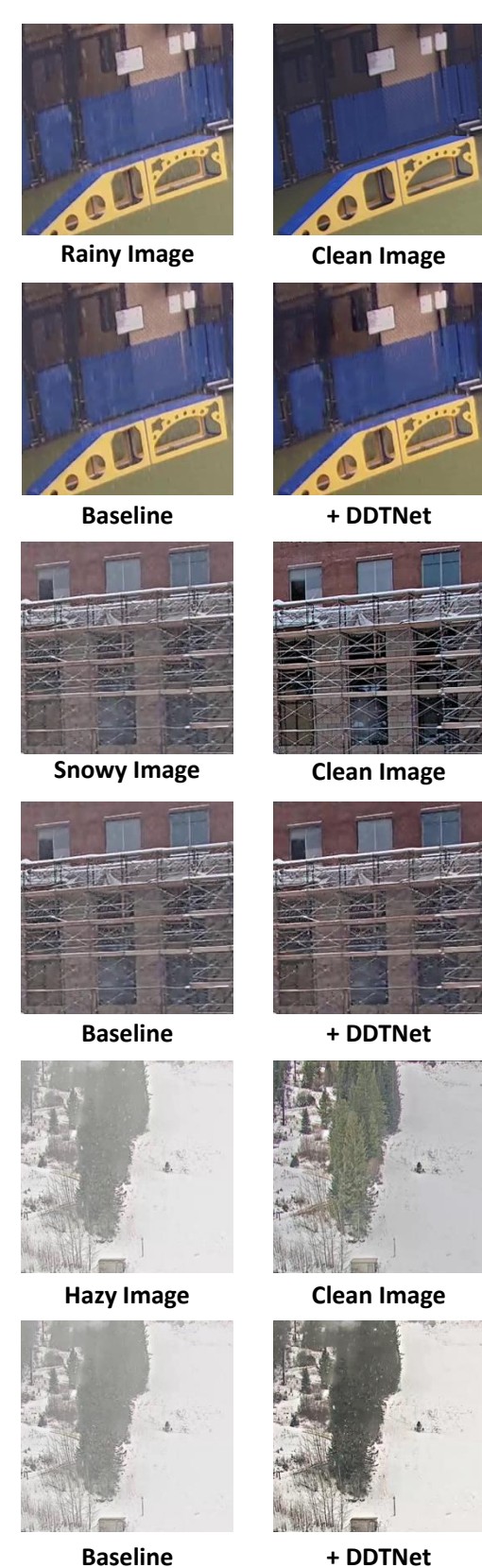

Figure A4: Qualitative comparison of AdaIR (Cui et al., 2025) on the WeatherStream (Zhang et al., 2023a) between its baseline and DDTNet-enhanced versions.

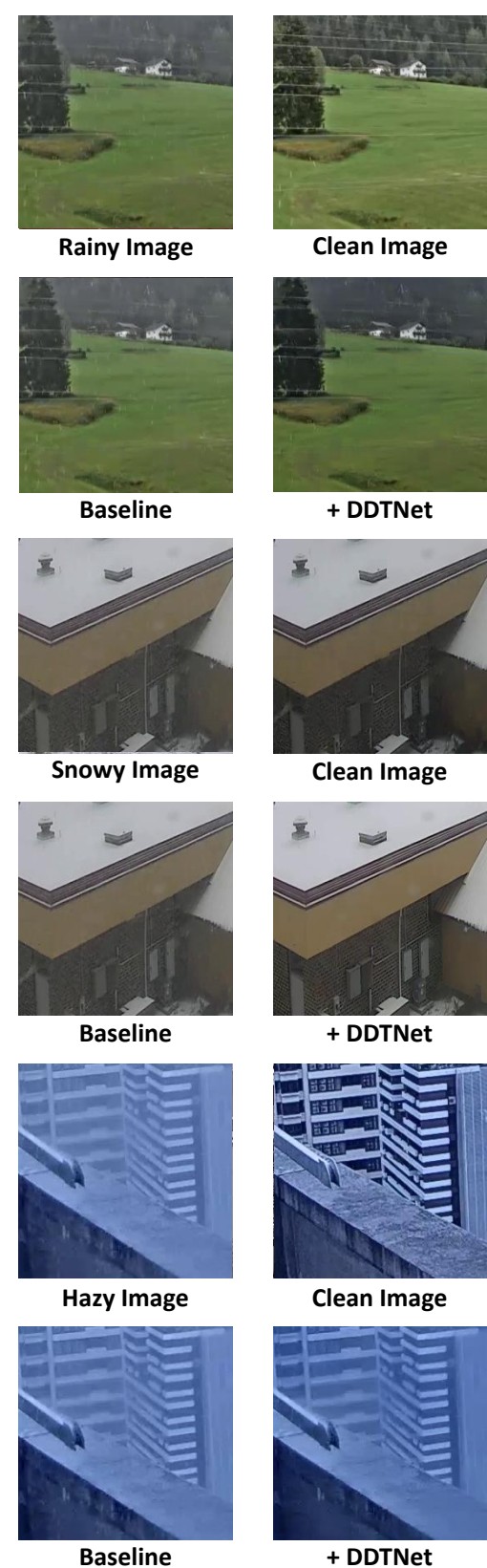

Figure A5: Qualitative comparison of DFPIR (Tian et al., 2025) on the WeatherStream (Zhang et al., 2023a) between its baseline and DDTNet-enhanced versions.

## A.3 THE USE OF LARGE LANGUAGE MODELS

We used large language models exclusively for linguistic refinement. All research motivations, ideas, methodologies, analyses, and results are entirely those of the authors. No AI tools were employed for conceptual development, data generation, analysis, or interpretation.

