# OpenReview forum: "DDTNet: Degradation Disentanglement and Transfer Network for Domain-Adaptive All-in-One Image De-weathering"
_ICLR.cc/2026/Conference — ICLR 2026 Conference Withdrawn Submission_

### Official Review · Reviewer_fH61 · 2025-10-24

**Soundness:** 2
**Presentation:** 1
**Contribution:** 2
**Rating:** 4
**Confidence:** 3

**Summary:**

DDTNet is a domain-adaptive all-in-one de-weathering framework that uses a Degradation Disentanglement Module (DDM) with two-stage Degradation-Coupled Attention (DCA) to disentangle weather patterns from a target degraded image and transfer them onto a source clean image, thereby synthesizing domain-adaptive paired data for fine-tuning arbitrary restoration models.

**Strengths:**

1.  Proposes a plug-and-play unpaired adaptation framework (DDTNet). The two-stage DCA explicitly separates content/degradation at the feature level and transfers target-domain degradations to source clean images to form fine-tuning pairs. The design is easy to integrate with existing all-in-one models.
2. Shows consistent gains on real/mixed weather conditions.

**Weaknesses:**

1.	Limited baselines & scope. The compared experiment is very limited, lacking prior domain adaptation approaches, unsupervised restoration methods, and representative supervised methods.
2.	No qualitative comparison vs Noise-DA. While quantitative results are provided, no visual comparisons are shown to illustrate behavioral differences.
3.	No evaluation on few-shot robustness. Experiments implicitly assume access to the full target domain. It remains unclear how DDTNet performs with few target-domain degraded images or under stricter data limits.
4.	Causality of gains not isolated. Lacks a control with equal number of pseudo pairs but without the proposed method to verify that improvements are not primarily from data augmentation alone.
5.	Presentation quality. The paper’s presentation is weak (low information density); visual comparisons serving the same purpose are scattered across multiple figures, hindering readability.

**Questions:**

Please refer to the weakness.

---

### Official Review · Reviewer_wodN · 2025-10-26

**Soundness:** 2
**Presentation:** 4
**Contribution:** 2
**Rating:** 4
**Confidence:** 4

**Summary:**

The paper proposes a domain adaptation model for all-weather restoration that produces degraded-clean image pairs containing the unknown degradation pattern present in the test image. Existing methods can then be fine-tuned on these pairs, enabling out-of-domain restoration. The proposed approach yields significant improvements in performance on unseen degradation patterns.

**Strengths:**

1. The paper is well written and easy to follow.
2. The idea is straightforward and the proposed components are intuitive.
3. DDTNet can be paired with any restoration network.
4. Fine-tuning on DDTNet generated degraded images improves out-of-domain performance.

**Weaknesses:**

1. Missing comparison with related work: [1] proposes a domain translation framework for multi-weather restoration which generates images under multiple weather conditions to extract content features. This work is quite similar and is not discussed or compared with.
2. Lack of ablations to show whether degradations are actually disentangled: The authors claim to disentangle degradation from clean content using the DCA module. While $I_o$ has the degradation in $I_d$, it is unclear whether the degradation was disentangled from the content features during the transfer. To demonstrate this, can the authors plot t-SNE of $\tilde{X}$ (Eqn. 6) for different degradations to show that they are separated? Additionally, a similar experiment can be performed for the outputs of the Content Encoder as well (in this case, I would not expect separation as content features would be clean).
3. Can DDTNet work for unseen and mixed degradations? For instance, can it generate pairs for unseen and mixed degradations which enable better restoration performance in these scenarios (eg. on Raindrop [2] or CDD testsets [3])?
4. Poor quality in qualitative results: In Fig. 6, the results for Snow and Haze seem to have many artifacts even after restoration using DDTNet. There is also a loss of structural detail compared to the ground truth. Is there any particular reason for this?

[1] Patil, Prashant W., et al. "Multi-weather image restoration via domain translation." Proceedings of the IEEE/CVF International Conference on Computer Vision. 2023.

[2] Qian, Rui, et al. "Attentive generative adversarial network for raindrop removal from a single image." Proceedings of the IEEE conference on computer vision and pattern recognition. 2018.

[3] Guo, Yu, et al. "Onerestore: A universal restoration framework for composite degradation." European conference on computer vision. Cham: Springer Nature Switzerland, 2024.

**Questions:**

1. In Fig. A2, how were the degradation features obtained?

For other questions see Weaknesses.

---

### Official Review · Reviewer_y9ct · 2025-10-30

**Soundness:** 2
**Presentation:** 2
**Contribution:** 2
**Rating:** 2
**Confidence:** 5

**Summary:**

This paper proposes DDTNet, a domain adaptation framework for all-in-one adverse weather image restoration. The method disentangles and transfers degradation patterns to generate domain-adaptive pairs, improving restoration under new conditions. Experiments show consistent gains on deraining, desnowing, and dehazing tasks.

**Strengths:**

1. The paper addresses a relevant problem in all-in-one image restoration and identifies domain gap issues that are indeed important for practical deployment.
2. The overall paper is well organized, and the experimental results are presented in a clear and structured manner.

**Weaknesses:**

1. The paper tackles both suboptimal cross-task performance and domain gaps, but it fails to clearly explain how these two issues are interconnected. The proposed solution addresses them separately, and it is unclear why they require a unified approach.
2. The paper describes a two-stage attention mechanism, where the first stage uses degradation-sensing tokens to retrieve degradation features, and the second stage uses image tokens to aggregate information from the degradation tokens. However, the significance of introducing the second stage is unclear, as the degradation features are already captured in the first stage.
3. Figure 5 only includes results from the proposed method. Since the paper claims to address the issues faced by prompt-based methods, particularly the suboptimal performance due to joint multi-task training, it would be more convincing to include comparisons with other prompt-based methods.
4. The paper mentions that the joint multi-task training strategy often leads to suboptimal performance on individual tasks, with noticeable performance drops compared to single-task models. However, it is unclear how the proposed DDTNet effectively addresses this issue, as it still appears to follow a similar multi-task prompt-based approach. How DDTNet mitigates the typical challenges of multi-task learning, particularly the degradation in performance on individual tasks.

**Questions:**

Please see above weaknesses.

---

### Official Review · Reviewer_GpWn · 2025-10-31

**Soundness:** 3
**Presentation:** 3
**Contribution:** 2
**Rating:** 4
**Confidence:** 4

**Summary:**

The paper introduces DDTNet, a domain-adaptive all-in-one image de-weathering model that disentangles and transfers degradation patterns between domains. It designs a Degradation Disentanglement Module (DDM) using a two-stage Degradation-Coupled Attention (DCA) mechanism to separate content and degradation features. DDTNet transfers learned degradation tokens to generate pseudo-pairs for fine-tuning existing models (e.g., PromptIR, AdaIR) on real data. Experiments show improved performance and generalization on synthetic and real-world benchmarks.

**Strengths:**

1. This paper introduces degradation disentanglement as a bridge for domain adaptation, which is interesting for multi-weather restoration. The two-stage DCA alternates between degradation- and image-focused attention.
2. Comprehensive experiments show consistent gains across several backbones and datasets.
3. Ablations and visualizations confirm that DDM and DCA improve both performance and interpretability.
4. Clear pipeline diagrams and equations; module functions are well defined.

**Weaknesses:**

1. Lacks a theoretical or analytical justification for the convergence or stability of the two-stage attention process. It remains unclear how DCA avoids information leakage between degradation and content streams.
2. The model contains ~31 M parameters and 33 ms inference latency on 256×256 inputs, but no scaling study is provided for high-resolution (1–2 K) deployment. The fine-tuning step per domain adds computational cost not fully quantified.
3. Although DDTNet claims domain adaptation benefits, no quantitative domain alignment metrics (e.g., FID, feature distance, or token separability) are reported beyond t-SNE visualization (Figure 5).
4. Experiments are confined to weather degradations (rain, snow, haze). It remains uncertain whether the framework generalizes to other degradation domains (e.g., low-light, noise, blur).
5. The baseline (Noise-DA) is primarily diffusion-based, while DDTNet is a transformer-style encoder–decoder. A direct comparison may conflate architecture effects with domain adaptation gains.

**Questions:**

1. How stable are the learned degradation-sensing tokens across different domains or seeds? Could the authors visualize token attention maps or show cosine similarity distributions?
2. If DDTNet is trained on rain/snow/haze, can it adapt to unseen degradation types (e.g., sandstorm, night haze) without retraining?
3. Beyond t-SNE, can the authors report feature alignment metrics (FID, MMD, or KL divergence) to substantiate domain adaptation claims?
4. How sensitive is performance to the number of fine-tuning epochs and data pairs? Could DDTNet support online adaptation (incremental updates)?
5. The degradation-sensing token number (M = 256) seems empirically chosen. Have the authors tested smaller or adaptive token counts for lightweight deployment?
6. If DDTNet is trained alongside PromptIR, can the same model be reused to fine-tune AdaIR without retraining DDTNet itself?
7. Could the proposed degradation disentanglement be extended to combined low-level restoration (e.g., deblurring + denoising + dehazing)?

---

### Note · Authors · 2025-11-12

I have read and agree with the venue's withdrawal policy on behalf of myself and my co-authors.